# Proton irradiation-decelerated intergranular corrosion of Ni-Cr alloys in molten salt

Weiyue Zhou [1], Yang Yang[2], Guiqiu Zheng[3], Kevin B. Woller [1], Peter W. Stahle[1], Andrew M. Minor [2,4] & Michael P. Short [1✉]

The effects of ionizing radiation on materials often reduce to "bad news". Radiation damage usually leads to detrimental effects such as embrittlement, accelerated creep, phase instability, and radiation-altered corrosion. Here we report that proton irradiation decelerates intergranular corrosion of Ni-Cr alloys in molten fluoride salt at 650 °C. We demonstrate this by showing that the depth of intergranular voids resulting from Cr leaching into the salt is reduced by proton irradiation alone. Interstitial defects generated from irradiation enhance diffusion, more rapidly replenishing corrosion-injected vacancies with alloy constituents, thus playing the crucial role in decelerating corrosion. Our results show that irradiation can have a positive impact on materials performance, challenging our view that radiation damage usually results in negative effects.

[1] Department of Nuclear Science and Engineering, Massachusetts Institute of Technology, Cambridge, MA, USA. [2] National Center for Electron Microscopy, Molecular Foundry, Lawrence Berkeley National Laboratory, Berkeley, CA, USA. [3] Nuclear Reactor Laboratory, Massachusetts Institute of Technology, Cambridge, MA, USA. [4] Department of Materials Science and Engineering, University of California, Berkeley, CA, USA. ✉email: hereiam@mit.edu

It has been well established that radiation accelerates corrosion of structural materials in today's reactors[1], as in-core experiments[2] and accelerator studies[3] have shown. However, multiple competing mechanisms may be present[4]. An overall increase in corrosion rate does not necessarily mean that each mechanism at play accelerates corrosion. The mechanisms and their relative strengths will necessarily change in different working fluids. In fact, observations of structural materials from the Molten Salt Reactor Experiment showed low corrosion rates[5]. This suggests that the influence of irradiation on corrosion in molten salts might be different from that in water-based systems. However, the reason(s) for this potential synergy have remained hidden for more than half a century.

Since corrosion products and normally protective oxides are soluble in the salt, oxide-based passivation does not occur, making corrosion in molten salt different from aqueous corrosion[6]. The predominant mechanism of corrosion of Ni-based alloys in molten fluoride salts in a practically useful sense, such as the Ni-20Cr alloy in this study, is selective dissolution of Cr (here the most susceptible element) into the salt[7]. The net loss of atoms resulting from dissolution-based corrosion results in voids containing the salt[8]. The distribution of voids is often localized, preferentially occurring along grain boundaries (GBs)[9]. One would therefore think that irradiation should increase the loss of Cr atoms by virtue of radiation-enhanced diffusion of Cr towards the material/salt interface. Indeed Bakai et al. have observed a considerable increase in Ni–Mo corrosion in NaF-ZrF$_4$ salt under electron irradiation[10].

Elucidating synergies between radiation and corrosion is one of the most challenging tasks impeding the deployment of advanced reactors[11,12], stemming from the combined effects of high temperature, corrosive coolants, and intense particle fluxes[13]. Here we show that proton irradiation can decelerate corrosion of Ni–Cr alloys in molten salt, contradicting radiation accelerated corrosion observed in liquid lead[14], molten salt[15], and water-based environments[1] while corroborating others[16]. The deceleration is confirmed and quantified by cross-sectional analysis revealing the depth of corrosion-induced voids. To explore the contribution of possible interactions, we first show that proton irradiation actually renders the salt more corrosive by a comparative experiment showing radiation accelerated corrosion in pure Fe. Therefore, a deceleration mechanism has to exist for the case of Ni–Cr alloys, which we propose to be the coupling between radiation-enhanced diffusion and corrosion-driven fluxes.

## Results

### Evidence of radiation-decelerated corrosion of Ni–Cr alloys.
Thin foil samples of Ni-20Cr exposed to 650 °C fluoride salt were irradiated with a beam of protons in a previously constructed, simultaneous irradiation/corrosion facility[17]. The central region of each sample was exposed to both protons and molten salt, while the outer region of each sample was exposed only to molten salt. Our experiments show that in all cases the unirradiated region suffers severe corrosion through the thickness of the foil, as evident by the penetration of salt to the other side. However, this barely occurs in the irradiated region. Figure 1a presents a schematic of our molten salt corrosion experiments undergoing selected area proton irradiation, showing that the material incurs proton damage without significant hydrogen implantation (Fig. 1b). The beam-facing side of the samples (Fig. 1e, h, k) can be distinctly divided into two regions, whose boundaries match the beam perimeter. Backscattered scanning electron microscope (SEM) images with corresponding elemental dispersive x-ray spectroscopy (EDX) point spectra (See Supplementary Fig. 1)

reveal the cause of the color difference to be the existence of the salt along the GBs on the outer (unirradiated) region of each sample. The region without irradiation (Fig. 1f, i, l) has salt decorating the GBs, while the irradiated region (Fig. 1d, g, j) is almost free of salt. This distinct difference in the two regions remains at the three different beam current densities tested, including 0.3, 0.4, and 0.5 µA cm$^{-2}$ corresponding to 0.018, 0.022, and 0.028 peak displacements per atom, respectively. The proton irradiation, as the only difference between these two regions, is shown to be the reason for the slower molten salt penetration.

### Quantification of deceleration by cross-sectional analysis.
To further confirm this effect, we ion-polished (Fig. 2a) each sample to expose cross-sections and performed SEM characterization. Figure 2c–h shows representative SEM images comparing the irradiated and unirradiated zones for different proton fluxes. In all cases, salt in the irradiated region rarely reaches the far side of the foil (Fig. 2c–e), while salt readily penetrates through the foil in the unirradiated region (Fig. 2f–h). The corrosion is quite localized in our experiments such that the depth of corrosion varies significantly at different locations. Therefore, we have performed a statistical analysis to reveal the distribution of corrosion depth. Our analysis process (Fig. 2i) starts from recording an ultra-high-resolution image in each region, representing an area of 1 mm along the y-axis and 30 µm along the x-axis. Afterwards, we divided the width of each image into more than 20,000-pixel rows, and calculated the maximum corrosion depth normalized by the sample thickness along each pixel row. To make quantitative comparisons, we plot the cumulative distribution function (CDF) of the normalized corrosion depth in Fig. 2j. Comparing the overall distributions of the irradiated and unirradiated regions at the same beam current, corrosion is clearly more severe in the regions without irradiation.

The wide distribution in corrosion depth is explained by both orientation-dependent attack and a mixture of simultaneous intergranular and transgranular corrosion. Here transgranular corrosion manifests as salt-filled voids penetrating transverse to GBs. Once initiated, these voids can grow in depth following the GBs, at the same time expanding to one or both sides of the adjacent grains. Thus, a more representative and quantitative distinction between the two sample regions can be made by comparing the extent of deepest attacks for each region, in effect comparing GBs possessing the most susceptible characters without directly measuring their orientation relationships. We therefore selected the 10% deepest attacks in each region (shown in the light red window in Fig. 2j) and calculated their averaged normalized corrosion depth. As shown in Fig. 2k, the irradiated region is corroded roughly two times less than the corresponding unirradiated region.

### Mechanistic insights from proton accelerated iron corrosion.
At this point there exist two possibilities to explain our observation of radiation-decelerated corrosion: (1) irradiation makes the alloy more corrosion-resistant; and/or (2) irradiation changes the chemistry of the molten salt so that it is less corrosive. To understand which one is valid, we have performed a comparative experiment on a pure Fe foil. Pure Fe corrodes much more uniformly in molten salts than a binary alloy because it contains no elements to selectively dissolve. Nonetheless the corrosion still results in surface asperities[18]. Corrosion etches valleys along GBs on the surface, which directly indicate the severity of the attack. We have collected a very large SEM composite image (26k by 26k pixels, 8 mm by 8 mm in size) of the salt-facing side of the pure Fe foil (Fig. 3b). Zoomed-in images from the edge (Fig. 3c) and the center (Fig. 3d) show that the center suffers from more severe

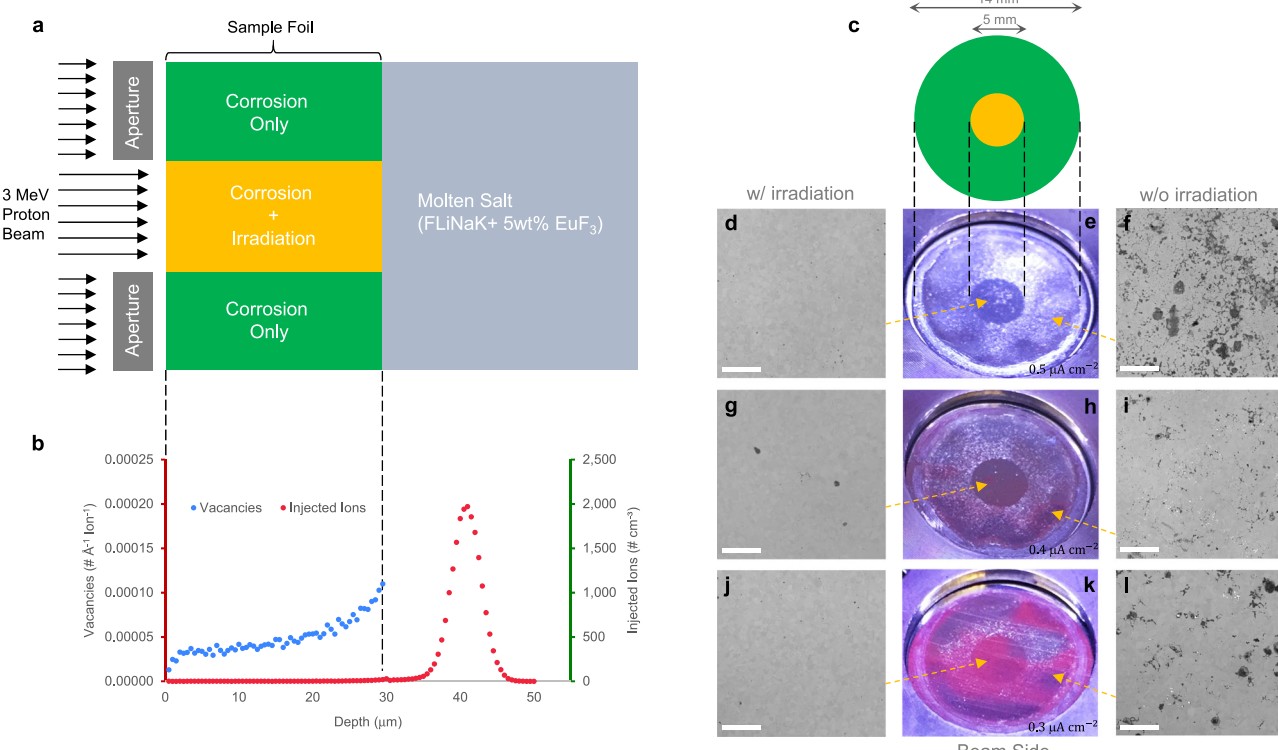

**Fig. 1 Schematics of the experimental setup, sample configuration, and beam-facing side comparison regions of Ni-20Cr samples. a** The experimental setup, showing how the two sample regions were created. **b** Distribution of primary radiation damage (blue dots) and proton deposition (red dots) along the irradiation direction for the Ni-20Cr corrosion experiments, simulated by SRIM[32]. **c** Schematic of the irradiated (in orange) and unirradiated (in green) zones. **d–l** Optical and representative SEM images of the beam-facing side of the Ni-20Cr foils after 4 h at 650 °C under various beam current densities. **d–f**, **g–i**, and **j–l** Correspond to beam current densities of 0.5, 0.4, and 0.3 μA cm⁻², respectively. Scale bar: 200 μm.

corrosion than the edge, implying that irradiation enhances corrosion in pure Fe in molten salts.

We then applied a machine-learning-based algorithm to partition the corroded and uncorroded regions (Fig. 3e). The corroded region (shown in red) does not overlap with the beam profile (white circle) very well, suggesting that local variation of corrosion severity in this experiment is caused by the interaction between the proton beam and the molten salt rather than the Fe foil itself. Irradiation renders the salt more corrosive, and this modified salt can flow away from the irradiation region. As a result, we also see some local radiation-free regions experiencing more severe corrosion. Our experiments on pure Fe indicate that proton irradiation accelerates corrosion of pure elements via increasing the corrosiveness of the molten salt. Thus, we confirm that the interaction between irradiation and the Ni–Cr alloy is the dominant cause of radiation-decelerated corrosion. One should note that the phenomenon of irradiation increasing the corrosiveness of the salt also exists in the case of Ni-20Cr corrosion. As it scales with increasing proton beam current, we would expect this effect to be more prominent at higher beam currents. Indeed, we found that as the beam current is increased, the overall corrosion attack of Ni-20Cr becomes more severe in both irradiated and unirradiated regions, as shown in Fig. 2k. Nonetheless, the irradiated regions of Ni-20Cr samples show significantly less severe corrosion attack than the unirradiated regions in all cases tested.

## Discussion

A model considering radiation-enhanced, bulk diffusion is proposed to explain how irradiation enhances corrosion resistance in

Fig. 4a, b. In our Ni–Cr binary alloy, Cr is preferentially depleted by molten salt because the redox potential of Cr is considerably lower than that of Ni in our LiF–NaF–KF+EuF₃ (FLiNaK+EuF₃) fluoride salt[19]. Cr in the bulk would diffuse out of the system via fast routes such as GBs to reach the salt[20]. The outward mass flux from GB to salt is compensated by the diffusion of lattice atoms (Ni and Cr) from the bulk to GBs, creating a self-healing mechanism to inhibit void formation along GBs. However, since bulk diffusion is much slower than GB diffusion at these temperatures, the vacancy density at GBs would increase during corrosion. As a result of corrosion, the atomic density of Cr at GBs would decrease, accompanied by an increase in Ni atomic density. Therefore, Ni is enriched and Cr is depleted along GBs (See Supplementary Fig. 2). One should note that the bulk diffusion of Cr and Ni to GBs does occur, but does not fully cease the increase of free volume at GBs. Eventually voids will nucleate at the GBs. Note that such bulk diffusion with or without irradiation is defined as:

$$D_{total} = D_i C_i + D_v C_v, \qquad (1)$$

where $D_i$ and $D_v$ represent interstitial and vacancy diffusivity, respectively, and $C_i$ and $C_v$ are site concentrations of interstitials and vacancies[21]. When there is no irradiation, the total diffusion flux of interstitials ($D_i C_i$) is negligible due to the very low concentration of thermal interstitials, typically at least 10⁶ times lower than the corresponding vacancy flux[22]. However, the presence of irradiation will change this.

Radiation damage cascades produce abundant interstitials within the grains in equal proportion to vacancies, which preferentially diffuse to defect sinks such as GBs[23]. Therefore, irradiation activates the interstitial term in Eq. (1), roughly doubling

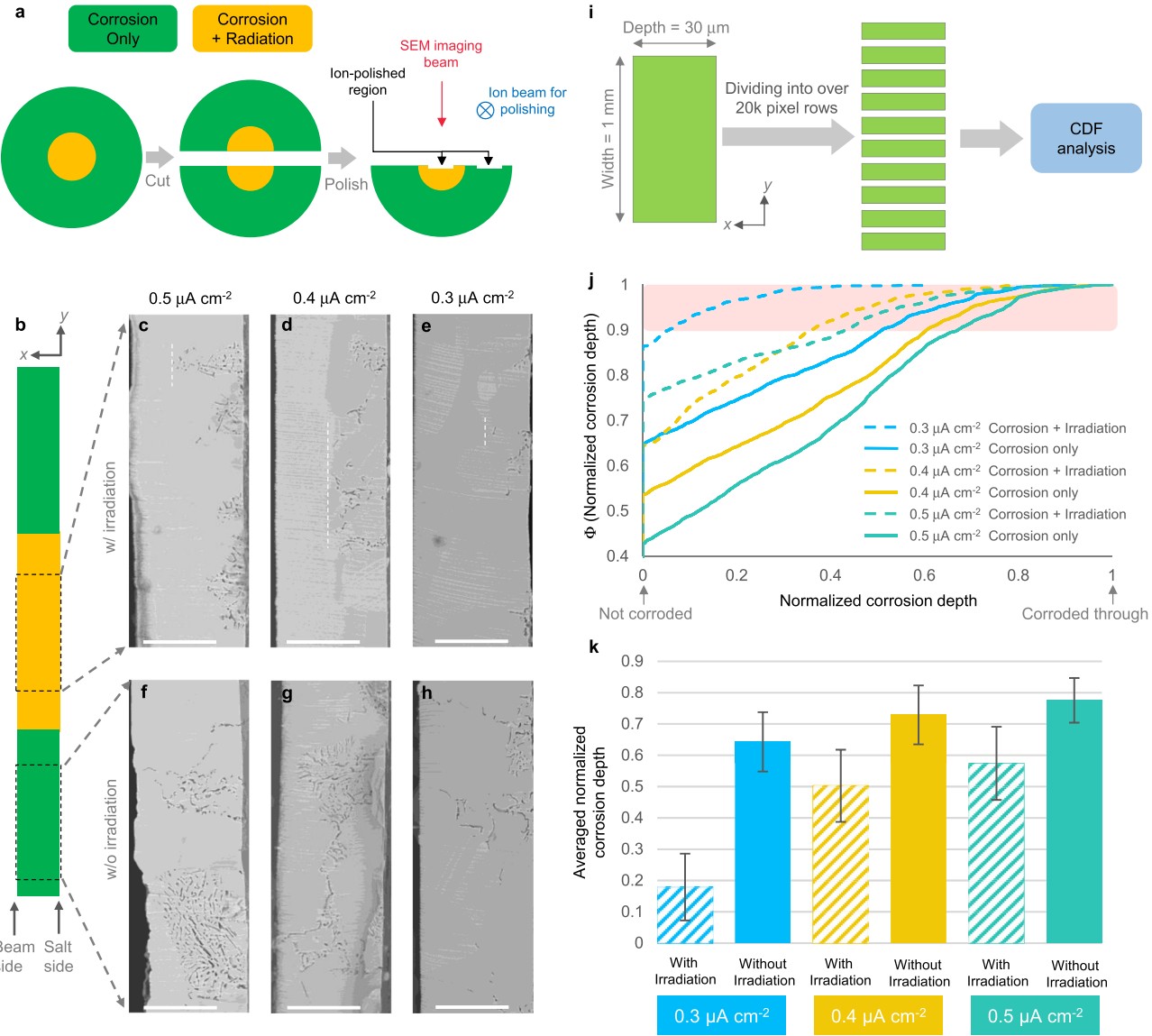

**Fig. 2 Cross-sectional comparison of corrosion with and without proton irradiation on Ni-20Cr. a** Ion-polishing process used to prepare samples for cross-sectional SEM imaging. **b** Schematic of the irradiated (in orange) and unirradiated (in green) zones for interpreting zones of SEM images. **c–h** Representative SEM images from different zones of Ni-20Cr samples under different beam current densities. Scale bar: 20 μm. **c–e** Show the irradiated zone, with white dashed-lines indicating the deepest attack. **f–h** Show the unirradiated zone. **i** Data analysis process for the Ni-20Cr foils. **j** Cumulative distribution function (CDF) of normalized corrosion depth, which is defined as corrosion depth normalized by sample thickness using data obtained by process in **i**. Dash lines represent data from the corrosion/irradiation region. Solid lines represent data from the corrosion only region. Blue, yellow, and dark green represent beam current densities of 0.3, 0.4, and 0.5 μA cm$^{-2}$, respectively. **k** Average, normalized corrosion depth using data from the 10% deepest corroded regions of each sample, as illustrated by the light red bar in **j**. Diagonal-filled bars represent data from corrosion/irradiation region. Solid bars represent data from corrosion only region. Meaning of colors is the same with **j**. Error bars denote one standard deviation.

the net flux of both Cr and Ni atoms towards GBs and thus accelerating the self-healing mechanism from its original, unirradiated rate. Because of that, void growth in grains adjacent to GBs will be much slower in the irradiated region than in the unirradiated region. In order words, irradiation enhances bulk diffusion and drives more atoms of both kinds from the grain to the GBs, where the Ni atoms will suppress void formation. Our model clarifies the mechanism of irradiation-decelerated intergranular corrosion to be enhanced mass transport to GBs via radiation-enhanced bulk diffusion. We emphasize that our model fundamentally differs from radiation induced segregation (RIS)[24], in that corrosion implies an open system while RIS assumes a closed system (See Supplementary Discussion). Our model is hypothesized to be generally effective in an alloy where one

element is preferentially dissolved by a corrosive fluid. Therefore, this effect should persist in other media, such as oxygen-poor molten lead, where selective dissolution is also the dominant mode of corrosion[25]. Recent evidence even suggests that proton radiation can decelerate corrosion of stainless steels in high temperature water[16], challenging conventional wisdom even further. In a more general sense, radiation has also been noted to improve mechanical properties of structural materials in certain circumstances[26].

Last, but not least, we compare the effects of proton and neutron irradiation on Ni-20Cr corrosion in molten salt. In our experiments, most protons stop in the salt, so the effect of injected hydrogen interstitials in the foil is negligible, though their effect on salt corrosiveness is strong. Unlike proton beams,

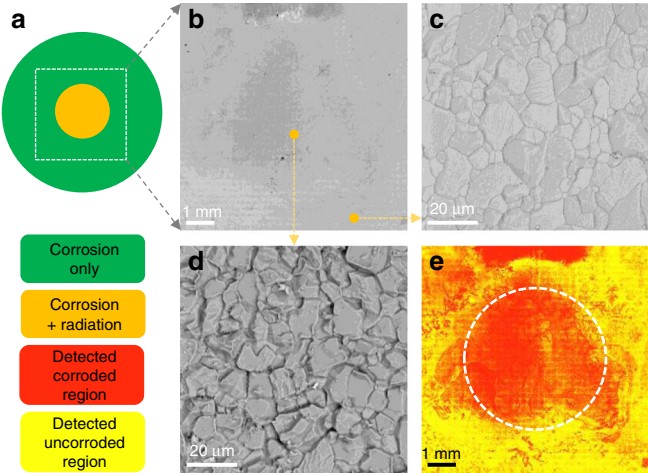

**Fig. 3 Salt-facing side of pure Fe following simultaneous corrosion/irradiation. a** Location on the Fe foil used for SEM imaging. Orange represents irradiated region. Green represents unirradiated region. **b** SEM image of the salt-facing side after 6 h at 650 °C under 0.4 µA cm$^{-2}$ proton irradiation. Scale bar: 1 mm. **c** Enlarged SEM image for the edge (unirradiated) region of the foil. Scale bar: 20 µm. **d** Enlarged SEM image for the center (irradiated) region of the foil. Scale bar: 20 µm. **e** Machine-learning based segmentation of **b**, showing auto-identified corroded (in red) and uncorroded (in yellow) regions. The white circle delineates the proton beam perimeter. Scale bar: 1 mm.

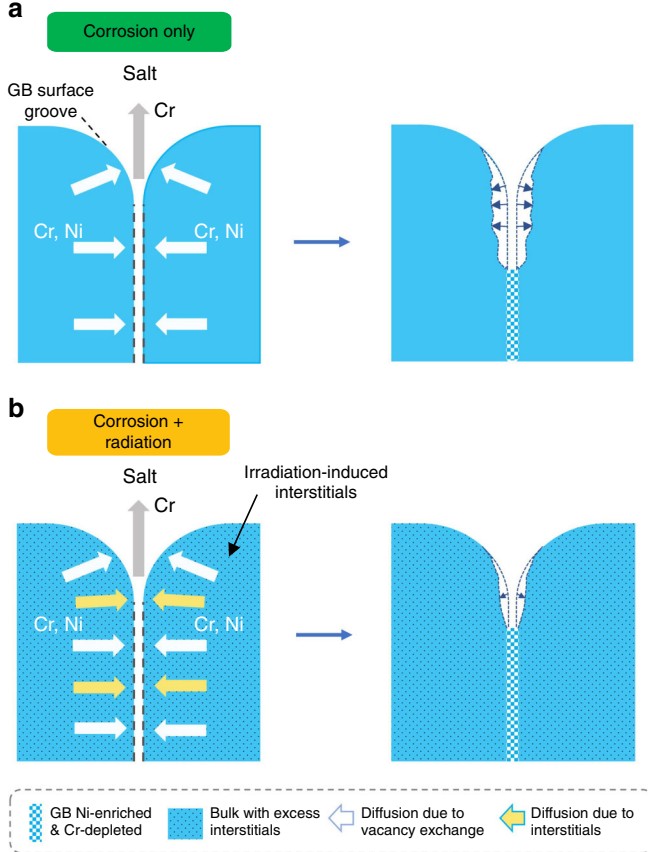

**Fig. 4 Proposed mechanism of radiation-decelerated corrosion. a** Schematic of the solid-state diffusion processes during molten salt corrosion in Ni-20Cr. **b** Schematic of the solid-state diffusion processes during molten salt corrosion in Ni-20Cr under the influence of proton irradiation.

neutrons will not interact with electrons directly and are not charged species themselves. Thus, we expect that salt undergoing neutron irradiation may be less corrosive than that undergoing proton irradiation, as neutron irradiation does not add charged chemical species to the salt (save for a small amount from transmutation production of tritium). As such, we expect that the irradiation-decelerated intergranular corrosion of Ni–Cr alloys under neutron irradiation to hold true, and perhaps even manifest a stronger deceleration compared to an equivalent flux of protons. Fortunately our experiments incur a similar particle flux as would be found in fast reactors, though the difference in particle type still raises the question of whether this self-healing mechanism will persist and dominate in nuclear reactors. However, our results reveal an encouraging mechanism of irradiation slowing down intergranular corrosion via enhanced bulk diffusion, which has important implications for the rapid development and down-selection of structural materials to finally usher in advanced nuclear fission and fusion reactors.

## Methods

**Salt and material preparation.** The FLiNaK+EuF$_3$ salt used for this work was produced in a temperature-controlled furnace housed inside an argon atmosphere glove box. Oxygen and moisture in the glove box were controlled to remain below 1 ppm. Powders of 99.99% pure LiF (CAS number 7789-24-4), 99.99% pure NaF (CAS number 7681-49-4), 99.99% pure KF (CAS number 7789-23-3), and 99.98% pure EuF$_3$ (CAS number 13765-25-8) were purchased from Alfa Aesar. LiF, NaF, and KF powders were melted into separate discs at 1000 °C in glassy carbon crucibles supplied by HTW Germany. EuF$_3$ powders were baked at 1000 °C. The salt discs of LiF, NaF, and KF were then broken into small pieces and measured by weight corresponding to the composition of a FLiNaK eutectic mixture (LiF–NaF–KF (46.5–11.5–42 mol%)). Baked EuF$_3$ powder was then added corresponding to be 5 wt.% of the total. Addition of EuF$_3$ increases the overall corrosion rate by increasing the redox potential of the salt[27]. EuF$_3$, when reacting with Cr, becomes EuF$_2$, which leaves no deposition on the sample surface. Afterwards, the salt mixture was melted inside another glassy carbon crucible by holding at 700 °C for 12 h. The final salt disc was then broken into small pieces. It was measured and melted into pellets roughly 3.5 g in weight. These pellets comprised the salt loaded into the corrosion/irradiation experiment.

An 80Ni-20Cr wt.% model alloy was made by Sophisticated Alloys Inc. in the form of 0.5-mm-thick sheet with a certified purity level of 99.95%. Then 30-µm-thick foils of this alloy were rolled by the H. Cross Company. All samples studied in this work consisted of 14-mm-diameter discs sectioned from the same rolled foil. Sample surfaces were smooth enough for the corrosion experiment without additional polishing.

For the pure Fe corrosion/irradiation experiment, a 99.5% (metals basis) pure Fe foil was purchased from Alfa Aesar with a thickness of 25 µm and used as-is.

**Simultaneous irradiation and corrosion experiments.** Details of the facility used in this study can be found in this reference[17]. Salt and sample loading were both performed in the same glove box mentioned previously. Then the vacuum-tight assembly was transferred out of the glove box to be connected to the proton accelerator beam line. The CLASS (Cambridge Laboratory for Accelerator Science) 1.7 MV Tandetron was used to accelerate protons to 3.0 MeV at the beam current densities listed in Fig. 1, measured with a Faraday cup. After the pressure reached 10$^{-6}$ Torr, the heater was started to reach 400 °C for at least half an hour, or until the pressure once again dropped below 10$^{-6}$ Torr. This ensured a proper bake-out of the as-assembled experimental facility. The temperature was increased to 650 °C over a period of roughly 1 h. Then the proton beam with controlled energy, beam shape, and current was introduced to the sample by withdrawing the Faraday cup. The beam current during the experiment was controlled to be between ±5% of the target beam current. Temperatures of the control thermocouple were maintained at ±1 °C. The two sides of the sample foil shared the same level of vacuum pressure during the experiments. Once the targeted duration of the experiment was reached, the beam was cut off, and the heater was stopped. It took less than half hour to cool down below 450 °C. During the entirety of the experiments, the pressure of the beam line was actively maintained between 10$^{-5}$ and 10$^{-6}$ Torr. After the system reached room temperature, the assembly was backfilled with ultra-high purity argon. Then it was disconnected from the beam line, transferred into the glove box, and disassembled.

**SEM analyses.** Prior to removal from the corrosion cell, pictures of the beam-facing side of the foils were taken by a camera, shown in Fig. 1e, h, k. Then the foils were cut out along their outer edge. The beam-facing side of each foil were imaged by a Phenom XL SEM in backscattering mode. These are shown in Fig. 1d, f, g, i, j, l. Cross-sections were obtained by the method shown in Fig. 2a using a JEOL SM-

Z04004T argon ion polisher with a voltage of 6 kV and a beam current between 150 and 180 μA. A cross-section wider than 1 mm was obtained by polishing for over an hour. This was repeated for all Ni-20Cr samples, and representative SEM images of each are shown in Fig. 2c–h.

**Cross-sectional image analysis of Ni-20Cr foils**. A grid of SEM images of each post-corrosion sample with the same magnification were taken in sequence, comprising an area 1-mm long. Grid stitching was performed using the stitching plugin[28] in Fiji[29]. Then the stitched image was straightened and cropped corresponding to 1 mm in width. Each image was then binarized with the auto threshold function of Fiji. For different sample foils, the contrast and brightness of the SEM images were slightly different. Thus, the auto threshold method was chosen so that the black pixels fully represented the salt-containing voids in the original SEM images. At the same time, patterns and artifacts from cross-section polishing along the argon beam-facing side of the foil were contrast-enhanced to become black pixels. These artificial black pixels were removed by comparing to the original SEM images. Then, along each pixel row perpendicular to the sample surface, the black pixel farthest away from the salt-facing side was detected and marked as the corrosion depth for this pixel row. These data comprise the graph in Fig. 2j.

**Image analyses of the salt-facing side of the Fe foil**. After the corrosion/irradiation experiment, the pure Fe foil was soaked in deionized water for over 24 h to remove the salt attached to its surface. Then the salt-facing side of the Fe foil was imaged with the same SEM. In total, 8 mm by 8 mm areas corresponding to 37 by 37 tiles of images were acquired. Then the 1369 tiles were stitched using the stitching plugin[28] in Fiji[29]. As the corroded region exhibited preferentially etched GBs while inside the grains remained clean, a simple color threshold method could not partition the corroded and uncorroded regions for the Fe sample after molten salt corrosion. Therefore, a machine-learning based image segmentation method, called Trainable Weka Segmentation[30], was used. First, a small part of the image containing both corroded and uncorroded regions was selected. Its size was 1/729th of the total area, and it was used to train a classifier model. Some of the regions were manually labeled as a training data set, the training and manual labeling process was iteratively performed several times until the classifier could correctly label all regions in the selection. Then the trained model was used to segment the entire high-resolution image on a workstation with 512 GB RAM, which took about 8 h.

**Transmission electron microscopy (TEM) analysis of corroded and irradiated GBs**. TEM specimens were prepared by $Ga^+$ focused ion beam lift-out from a cross-section of each foil. The beam energy used for thinning the sample was gradually reduced from 30 to 2 keV. STEM-EDX characterization was performed using an FEI ThemIS TEM at the national center for electron microscopy (NCEM) in the Molecular Foundry of Lawrence Berkeley National Laboratory. The TEM was operating in STEM mode with an electron beam energy of 300 keV. Linescans of Ni and Cr intensity were acquired across selected GBs to obtain evidence of Ni enrichment and Cr depletion.

## Data availability

All the data from this study, including original micrographs, processed data, scripts, and compiled results are freely available at the GitHub repository for this paper[31].

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

## Acknowledgements

The authors gratefully acknowledge funding from the Transatomic Power Corporation under Grant No. 023875-001, and the US Department of Energy Nuclear Energy University Program (NEUP) under Grant No. 327075-875J. A.M. acknowledges the support of FUTURE (Fundamental Understanding of Transport Under Reactor Extremes), an Energy Frontier Research Center funded by the U.S. Department of Energy, Office of Science, Basic Energy Sciences. Y.Y. was supported by the Director, Office of Science, Office of Basic Energy Sciences, Materials Sciences and Engineering Division, of the U.S. Department of Energy under Contract No. DE-AC02-05-CH11231 within the Mechanical Behavior of Materials (KC 13) program at the Lawrence Berkeley National Laboratory. The authors acknowledge support by the Molecular Foundry at Lawrence Berkeley National Laboratory, which is supported by the U.S. Department of Energy under Contract No. DE-AC02-05-CH11231. The authors wish to thank Charles Forsberg (MIT), Peter Hosemann and Raluca Scarlat (UC Berkeley), Gabriel Meric de Bellefon

(Kairos Power), En-Hou Han (IMR, China), and Il-Soon Hwang (UNIST, Korea) for discussions in guiding this study. Thanks are due to Mitchell Galanek, Ryan Toolin, Ed Lamere, and Ryan Samz from MIT's Environmental Health and Safety (EHS) department for verifying device safety and shielding, and to Amy Tatem-Bannister and William DiNatale for training and access to the argon ion cross-section polisher.

## Author contributions

W.Y.Z. constructed the corrosion/irradiation facility with assistance from G.Q.Z. K.B.W., and P.W.S. W.Y.Z. conducted all corrosion and irradiation experiments and SEM characterizations. Y.Y. and A.M. performed TEM sample preparation and TEM characterization as well as machine-learning auto-identification of corroded regions on the pure Fe sample. M.P.S. conceived of the original project and oversaw its execution, providing regular guidance. W.Y.Z, Y.Y, and M.P.S wrote the paper. All authors contributed to the analysis of the results.

## Competing Interests

The authors declare no competing interests.
