## [Peer Review File · Nature Communications]

Reviewers' Comments:

Reviewer #1:

Remarks to the Author:

This paper studies the synergies between radiation and corrosion of a Ni-Cr alloy and reports that proton irradiation decelerates intergranular corrosion of this alloy in molten fluoride salt at 650C. The authors designed an excellent experiment and make it possible to study the radiation-corrosion synergies of a nuclear material using an ion beam facility. Further, the results they obtained are very interesting and the discussions are reasonable. It is a well written manuscript and suitable to be published in this journal.

Suggestions:

Caption of Figure 2, page 9: "Cross-sectional comparison of corrosion with and without molten salt irradiation on Ni-20Cr." should be corrected as "Cross-sectional comparison of corrosion with and without proton irradiation on Ni-20Cr."?

Reviewer #2:

Remarks to the Author:

This paper describes the results of proton irradiation of a Ni-20Cr alloy in contact with FLiNaK to assess the effect of irradiation on corrosion of the alloy. The topic is timely and important as it addresses the concern that irradiation may significantly accelerate corrosion in molten salt, as it has been found to do with Zircaloy in light water reactors. The result here indicates that in fact, irradiation decelerates corrosion and the authors offer an explanation for that observation. The results are novel and the paper is well written and easy to follow. The statistical analysis is appropriate. The supplemental information is important and useful. Results are clearly presented and the analysis is easy to follow. Nevertheless, there are several points that the authors need to address, in particular the role of irradiation- and corrosion produced-defects.

1) The first concern I have is that the seal between the foil and the molten salt may not be sound. That is, a compression seal is prone to leakage and if this occurs, it could explain the corrosion pattern on the Fe foil as well as salt on the vacuum side of the Ni-20Cr foil. What evidence do the authors have that indicates that the seal did not leak? What was the pressure in the beamline for this experiment?

2) P. 2, Line 100: The authors state that corrosion is a balance between irradiation-increased corrosiveness of the salt and deceleration of corrosion due to radiation effects in the alloy. They then deduce that that increased corrosion with beam current density is because of increased corrosion attack due to increased corrosiveness of the salt under irradiation. But their mechanism (described later) predicts increased corrosion deceleration with beam current density, and as they state on line 95, this process "outperforms" the corrosiveness of the salt. So, by their own logic, the corrosion rate should decrease with current density, not increase.

3) P. 3, line 58: This statement essentially says that the introduction of charged hydrogen ions in molten salt makes a more corrosive environment. But no explanation is provided as to why this should occur.

4) P. 3, line 35 and Supplementary section: The authors argue that radiation enhanced diffusion accelerates the transport of Cr to the grain boundary, and that this is, in part, responsible for deceleration of corrosion in molten salt. However, since GB diffusion is much faster (orders of magnitude) than bulk diffusion, the supply of Cr to the boundary simply serves to increase its dissolution rate - it doesn't raise the Cr level at the boundary since it leaves by dissolution once it arrives. Therefore, the RED can't explain the beneficial effect of irradiation. If RED transports Cr to the GB, and molten salt leaches Cr, then the corrosion should be worse under irradiation than

without it. In fact, the increase in corrosion with current density supports this.

They also argue that RIS should be insignificant in their alloy at 650°C. But then on p. 1, line 71 of the Supplement, they argue that RED is enhanced by RIS. This statement is false on two accounts. First, RIS cannot enhance RED. It is the other way around. Second, on line 43, they discounted RIS. So they cannot discount it and then come back and claim it has an effect.

Further, on line 80 they cite both RIS and corrosion-induced vacancy injection as being jointly responsible for the deceleration of corrosion of Ni-20Cr in molten salt. Again, having dismissed RIS as insignificant, it cannot now be cited as a major factor in the observed corrosion.

Their understanding of the role of irradiation enhancement of Ni and Cr supply to the grain boundary is also incorrect. In a Ni-Cr alloy, Cr diffuses preferentially against the vacancy flux. Ni diffuses easier via interstitials than does Cr, so the net result is Ni enrichment and Cr depletion at grain boundaries. The reason RIS is minimal at very high temperature is that the thermal vacancy concentration is such that a concentration gradient cannot be supported. RED does not change the preference of Cr to deplete and Ni to enrich. The injection of vacancies at the grain boundary can alter the direction of flow, but for that to happen, it would have to "overpower" the vacancy supply from irradiation. The authors have not done an adequate assessment of this situation to make conclusions on the role of vacancy injection due to corrosion. So their conclusions on its effect are really speculation.

Reviewer #3:

Remarks to the Author:

I do not think the paper can be published in the journal.

On page 1

1. Line 33 "radiation damage always results in negative effects", this statement is not right, some experiments have shown that radiation can have some positive effects on corrosion, even for neutron radiation
2. Line 52 "making corrosion in molten salt different from aqueous corrosion" actually, both types of corrosion in molten salt and aqueous solution are electrochemical process including anodic dissolution and cathodic reduction, the corrosion mechanisms are the same.
3. Line 53 "the predominant... , is selective dissolution of Cr" Actually, selective dissolution of Cr is just one type of the corrosion, which depends on the corrosion potential of the molten salt. if the corrosion potential is above the Ni dissolution potential, Ni can be dissolved very faster than Cr. Corrosion not only depends on the thermodynamic potential but also depends on the dissolution kinetics when the corrosion potential is above its dissolution potential
4. Line 69-71, "in liquid lead...oxidation was the primary corrosion mechanism". Actually, in liquid lead, the major corrosion mechanism is physical dissolution, not oxidation, the oxygen control technology is applied in liquid lead to reduce the corrosion by stopping the physical dissolution

On Page 3:

- 1) Eq.1 seems not right, the unit on the left and right are not the same, the authors need to double check the reference and actually understand what the equation is in the reference [25]
- 2) The authors think that it is EuF_3 that leads to corrosion. I highly doubt it. I read through the reference cited, the original reference was proposed to use $\text{Eu}^{3+}/\text{Eu}^{2+}$ couple to mitigate corrosion by decreasing corrosion potential. The author needs to make at least a very simple thermodynamic calculation to show that the standard potential of $\text{Eu}^{3+}/\text{Eu}^{2+}$ is more positive than Cr^{2+}/Cr , if it is more positive, which means it is possible that it is Eu^{3+} that leads to corrosion, or it is impossible
- 3) The impurity of the salt is unknown, especially, the oxygen level and the moisture level, both oxygen and moisture can be the corrosion sources. Therefore, the oxygen level and moisture level at which they prepared the salt must be given.

4) Figure 4 is not complete. The authors need to know that the molten salt corrosion includes two reactions: anodic dissolution/oxidation and cathodic reduction. The corrosion of Cr is through a reaction $\text{Cr} - 2\text{e}^- \rightarrow \text{Cr}^{2+}$, there must be a species that can be reduced to "eat" electron released by Cr dissolution. The species must be diffuse in the salt to reach the corrosion surface, the second reaction was completely neglected in Figure 4. I do not think the author really understand the corrosion mechanisms by molten salt.

Another important factor that will lead to corrosion was not addressed in the document. The author used proton (H^+) which is completely stopped in the salt and accumulate in the salt. H^+ is a corrosive species in salt because $\text{H}^+ + \text{e}^- \rightarrow \text{H}_2$ which can be reduction reaction that may lead to corrosion, which is not related to radiation. It is because the author artificially introduced corrosive species into their system. Therefore, H^+ influence on corrosion during experiments needs to be addressed and discussed.

NATURE COMMUNICATIONS

Proton Irradiation-Decelerated Intergranular Corrosion of Ni-Cr Alloys in Molten Salt

W.Y. Zhou, Y. Yang, G. Q. Zheng, K. B. Woller, P. W. Stahle, A. M. Minor, M. P. Short

April 23, 2020

To the Editors and Reviewers of Nature Communications,

Thank you for reviewing our manuscript entitled "Proton Irradiation-Decelerated Intergranular Corrosion of Ni-Cr Alloys in Molten Salt." We have carefully gone through all the comments and responded to them below. Corresponding changes are also marked in **red** in the revised manuscript.

Reviewer #1

- Comment 1): *Caption of Figure 2, page 9: "Cross-sectional comparison of corrosion with and without molten salt irradiation on Ni-20Cr." should be corrected as "Cross-sectional comparison of corrosion with and without proton irradiation on Ni-20Cr."?*

Thanks for catching this error. We have changed "molten salt irradiation" to "**proton** irradiation" at the corresponding position in the revised manuscript.

Reviewer #2

We greatly appreciate the insightful comments from Reviewer #2 to guide the revision. We agree with Reviewer #2 that the topic covered by the manuscript is timely and important, and for the recognition of our results being novel and the manuscript being clear. Now we will carefully address each comment from Reviewer #2. We separate comment #4 into four sub-comments to address individually.

- Comment 1): *The first concern I have is that the seal between the foil and the molten salt may not be sound. That is, a compression seal is prone to leakage and if this occurs, it could explain the corrosion pattern on the Fe foil as well as salt on the vacuum side of the Ni-20Cr foil. What evidence do the authors have that indicates that the seal did not leak? What was the pressure in the beamline for this experiment?*

The seal between the foil and the molten salt was indeed one of the biggest challenges we faced, and overcame, when designing the experimental setup. We would like to point out that the compression seal here is a liquid (molten salt)-tight seal, not a gas-tight seal. The whole corrosion cell sits inside another vacuum housing. Thus, there is little to no pressure difference across the sample/accelerator boundary. Also the corrosion pattern on the Fe foil we show is on the salt-facing side of the foil, not on the beam-facing side. In fact, the beam-facing side of the Fe foil showed no visible salt after the experiment under SEM. There is another case with a Ni-201 (commercially pure Ni) foil after a similar testing conditions, i.e., 4 hours at 650°C under 2.0 $\mu\text{A}/\text{cm}^2$ of proton irradiation. We have scanned through the beam-facing side of this Ni-201 foil, and have found no trace of the salt. The Ni-201 foil therefore represents the cleanest test of the seal quality, as it does not corrode at all in our FLiNaK salt, thus if any salt were visible on the beam-facing side it would have been due to a seal leak. Fortunately there was none.

There was another instance when we didn't apply force on the bolts evenly, and the liquid-tight seal was not formed successfully. The liquid salt leaked out from the corrosion cell, filling the gap between the corrosion cell and the cap. Therefore we do know what happens when the seal is not sound, and what a liquid salt leak looks like. None of the tests in the manuscript belong to this scenario. Therefore, We did not change the original manuscript regarding this important factor.

The pressure of the beamline for this experiment is continuously monitored to remain between 10^{-5} Torr and 10^{-6} Torr. To make this more readily accessible to readers, we added one sentence under “METHODS: B. Simultaneous irradiation and corrosion experiments” as: *During the entirety of the experiments, the pressure of the beam line was actively maintained between 10^{-5} and 10^{-6} Torr.*

- Comment 2): *P. 2, Line 100: The authors state that corrosion is a balance between irradiation-increased corrosiveness of the salt and deceleration of corrosion due to radiation effects in the alloy. They then deduce that that increased corrosion with beam current density is because of increased corrosion attack due to increased corrosiveness of the salt under irradiation. But their mechanism (described later) predicts increased corrosion deceleration with beam current density, and as they state on line 95, this process “outperforms” the corrosiveness of the salt. So, by their own logic, the corrosion rate should decrease with current density, not increase.*

Thanks for pointing this out. In fact, we were trying to express that the two ways which protons influence the corrosion of Ni-20Cr co-exist and act upon the foil at the same time. Therefore, the results in Figure 2 are from both types combined. We used the word “outperformed” to express that the material corrosion deceleration mechanism (described later) has to be stronger than the increased salt corrosiveness effect. Otherwise, one would observe an accelerated corrosion of Ni-Cr samples instead of decelerated corrosion. We were not trying to comment on the relationship between acceleration/deceleration and beam current density. In fact, we also noticed that with higher beam current density, the increased salt corrosiveness effect tries to overcome the material corrosion deceleration mechanism (described later). This can be observed by the relative ratios between irradiated and unirradiated corrosion rates in Figure 2k (ratio of the averaged normalized corrosion depth under irradiation over without irradiation), which approach unity with higher beam current density. However, we were not trying to express this point in the original manuscript as we felt the proof for this point is not strong enough. Here, the choice of word “outperformed” is not appropriate as the reviewer mentions.

The related sentences from the original manuscript are: *One should note that the phenomenon of irradiation increasing the corrosiveness of salt would also exist in the case of Ni-20Cr corrosion. However, it is outperformed by the radiation damage-decelerated corrosion in our experiments. As it scales up with increasing proton beam current, we would expect this effect to be more prominent at higher beam current. Indeed, we found that as the beam current is increased, the overall corrosion attack of Ni-20Cr becomes more severe in both irradiated and unirradiated regions, as shown in Figure 2k.*

These sentences have been modified to read: *One should note that the phenomenon of irradiation increasing the corrosiveness of salt also exists in the case of Ni-20Cr corrosion. As it scales with increasing proton beam current, we would expect this effect to be more prominent at higher beam currents. Indeed, we found that as the beam current is increased, the overall corrosion attack of Ni-20Cr becomes more severe in both irradiated and unirradiated regions, as shown in Figure 2k. Nonetheless, the irradiated regions of Ni-20Cr samples show less severe corrosion attack than the unirradiated regions in all cases tested.*

- Comment 3): *P. 3, line 58: This statement essentially says that the introduction of charged hydrogen ions in molten salt makes a more corrosive environment. But no explanation is provided as to why this should occur.*

Thanks for catching this. We are not yet sure about the detailed mechanism of increased corrosiveness by proton irradiation. As a consequence, we tread very carefully when introducing the results from the pure Fe foil, only focusing on the fact that the regions of accelerated corrosion switched positions compared to the Ni-20Cr foils. We only wanted to show that the proton irradiation was indeed making the salt more corrosive. The mechanism behind this increased corrosiveness is out of the scope of this manuscript. In fact, this phenomenon itself is a very intriguing topic to study. However, we were not careful enough on page 3 when we were trying to compare the difference between neutrons and protons. The way we wrote it gave the impression that the charged hydrogen ions are the mechanism behind the increased corrosiveness, when in reality we do not yet know the species responsible.

The original sentences are: *In our experiments, most protons stop in the salt, so the effect of injected hydrogen interstitials in the foil is negligible, though their effect on salt chemistry is strong. Unlike proton beams, neutrons will not introduce charged hydrogen when they stop in the salt. Thus, we expect that salt undergoing neutron irradiation may be less corrosive than that undergoing proton irradiation.*

These sentences have been modified to read: *In our experiments, most protons stop in the salt, so the effect of injected hydrogen interstitials in the foil is negligible, though their effect on salt corrosiveness is strong. Unlike proton beams, neutrons do not interact with electrons directly and are not charged species themselves. Thus, we expect that salt undergoing neutron irradiation may be less corrosive than that undergoing proton irradiation, as neutron irradiation does not add charged chemical species to the salt (save for a small amount from transmutation production of tritium).*

- Comment 4-1): *P. 3, line 35 and Supplementary section: The authors argue that radiation enhanced diffusion accelerates the transport of Cr to the grain boundary, and that this is, in part, responsible for deceleration of corrosion in molten salt. However, since GB diffusion is much faster (orders of magnitude) than bulk diffusion, the supply of Cr to the boundary simply serves to increase its dissolution rate - it doesn't raise the Cr level at the boundary since it leaves by dissolution once it arrives. Therefore, the RED can't explain the beneficial effect of irradiation. If RED transports Cr to the GB, and molten salt leaches Cr, then the corrosion should be worse under irradiation than without it. In fact, the increase in corrosion with current density supports this.*

The reviewer is correct in the sense that if radiation enhanced diffusion (RED) helps transport Cr to the grain boundary, *and one does not also consider the additional transport of Ni to the same boundaries*, then the Cr leaching rate could remain higher under irradiation. However, focusing only on the diffusion process of Cr neglects the crucial role of Ni. We explained the corrosion mechanism in the original manuscript on page 2, line 110: *The outward mass flux from GB to salt is compensated by the diffusion of lattice atoms (Ni and Cr) from the bulk to GBs, creating a self-healing mechanism to inhibit void formation along GBs.* The radiation enhances both Cr and Ni transport. Ni atoms, being inert at GBs, will slow the development of salt-containing voids. This was what we were trying to deliver.

To make it more clear, we have modified the relevant sentences on page 3, starting at line 21. The original sentences are: *Therefore, irradiation activates the interstitial term in Equation (1), doubling the net flux of atoms towards GBs and thus accelerating the "self-healing" mechanism from its original, unirradiated rate. Because of that, void growth in grains adjacent to GBs will be much slower in the irradiated region than in the unirradiated region. In order words, irradiation enhances bulk diffusion and drives more atoms from the grain to the GBs to suppress void formation.*

These sentences have been modified to read: *Therefore, irradiation activates the interstitial term in Equation (1), roughly doubling the net flux of both Cr and Ni atoms towards GBs and thus accelerating the "self-healing" mechanism from its original, unirradiated rate. Because of that, void growth in grains adjacent to GBs will be much slower in the irradiated region than in the unirradiated region. In order words, irradiation enhances bulk diffusion and drives more atoms of both kinds from the grain to the GBs, where the Ni atoms will suppress void formation.*

- Comment 4-2): *They also argue that RIS should be insignificant in their alloy at 650°C. But then on p. 1, line 71 of the Supplement, they argue that RED is enhanced by RIS. This statement is false on two accounts. First, RIS cannot enhance RED. It is the other way around. Second, on line 43, they discounted RIS. So they cannot discount it and then come back and claim it has an effect.*

Thanks for catching this. The sentence on page 1, line 71 of the supplementary materials is not correctly written. The original sentence was: *Radiation enhanced diffusion, however, is greatly enhanced in proportion to defect creation, both by RIS and by vacancy injection from the corrosion process.* This sentence is now revised as: *Diffusion, however, is greatly enhanced in proportion to defect creation, both by radiation and by vacancy injection from the corrosion process.*

- Comment 4-3): *Further, on line 80 they cite both RIS and corrosion-induced vacancy injection as being jointly responsible for the deceleration of corrosion of Ni-20Cr in molten salt. Again, having dismissed RIS as insignificant, it cannot now be cited as a major factor in the observed corrosion.*

Through the discussion in the supplementary materials, we came to the conclusion that the distribution profile we observed cannot *principally* be a result of RIS. Therefore, corrosion-driven fluxes were responsible for the vast majority of the observed Ni enrichment and Cr depletion profile. But we cannot completely discount RIS, as it stems from the preferential coupling between defect fluxes and atom fluxes. This preferential coupling will exist regardless of the cause of the defect fluxes. We argued that the coupling of both Cr and Ni to the defect fluxes, instead of the preferential coupling due to unequal atom-specific defect fluxes, is dominant in our case. However, the sentence on page 1, line 80 of the supplementary materials, was not well written to convey this message. We therefore revise it to make it more clear.

The original sentences are: *Thus it is likely that both RIS and corrosion-induced vacancy injection are jointly responsible for the deceleration of corrosion of Ni-20Cr in molten salt. The latter is shown to be the dominant process, both quantitatively from previous models/experiments and mechanistically by recognizing that irradiation-decelerated corrosion in molten salt is an open system with additional vacancy injection.*

They are now modified as follows: *Thus the preferential coupling between defect fluxes and atom fluxes can still occur in our experiments. However, the coupling of both Ni and Cr atoms to defect fluxes — against vacancy flux and follow interstitial flux — likely dominates over the preferential coupling. Therefore, radiation enhanced diffusion, coupled to the corrosion induced diffusion process, is recognized as the mechanism behind the deceleration phenomenon.*

- Comment 4-4): *Their understanding of the role of irradiation enhancement of Ni and Cr supply to the grain boundary is also incorrect. In a Ni-Cr alloy, Cr diffuses preferentially against the vacancy flux. Ni diffuses easier via interstitials than does Cr, so the net result is Ni enrichment and Cr depletion at grain boundaries. The reason RIS is minimal at very high temperature is that the thermal vacancy concentration is such that a concentration gradient cannot be supported. RED does not change the preference of Cr to deplete and Ni to enrich. The injection of vacancies at the grain boundary can alter the direction of flow, but for that to happen, it would have to “overpower” the vacancy supply from irradiation. The authors have not done an adequate assessment of this situation to make conclusions on the role of vacancy injection due to corrosion. So their conclusions on its effect are really speculation.*

We had the same description of the preferential coupling of Ni and Cr atoms to defect fluxes. The sentences are copied here (page 1, line 11 of supplementary materials): *RIS is often explained by Inverse Kirkendall (IK) theory, where different elements preferentially couple to either the interstitial flux or vacancy flux. The similarly directional fluxes of vacancies and interstitials towards GBs drive the elemental segregation resulting in RIS. In a Ni-Cr binary alloy system undergoing RIS, Ni is predicted and observed to be enriched, and Cr depleted.*

For the reasons that RIS is minimal at very high temperature, we agree that the concentration gradient cannot be supported. We also agree with reference [7] of the supplementary materials, which states “At high temperatures, a large thermal vacancy concentration leads to a high diffusion rate of alloying elements as well as to a high defect recombination rate; the latter reduces the defect fluxes to sinks, and hence, the amount of solute segregation, while the former increases the back diffusion of segregated alloying elements.” This was the reason we wrote in the supplementary materials: *... as it is well known that RIS subsides at higher temperatures due to chemical back-diffusion.* on page 1, line 40. To make it more clear, this sentence is now modified to read as: *... as it is well known that RIS subsides at higher temperatures due to chemical back-diffusion and high recombination rates of defects.*

We agree that RED does not change the preference of Cr to deplete and Ni to enrich. In fact, this is the reason we used one paragraph (starting from line 25 in the supplementary materials) to show that the observed Cr depletion and Ni enrichment cannot be credited primarily to RIS. Therefore, another mechanism must be responsible.

Under irradiation, it is difficult to estimate which is higher, the injection of vacancies from the grain boundary due to corrosion, or the flow of vacancies generated by irradiation to the grain boundary. Nonetheless, we know these two defect fluxes oppose each other. The generation of vacancies by irradiation inside grains will decrease the flux of vacancy injection due to corrosion. From another point of view, vacancy flux is the reverse of atom flux (if we skip the preferential coupling here). In other words, radiation-generated vacancies will decrease the tendency of atom transport due to vacancy exchange. Despite the decrease of the driving force for vacancy migration being clear, the actual speed of vacancy atom exchange is difficult to estimate since the vacancy diffusion coefficient might also be altered.

It is critical to compare an open and closed system to better understand in this case. In a closed system, for example RIS at a grain boundary, both interstitials and vacancies flow towards the grain boundary since they need to annihilate at the grain boundary. Otherwise the grain boundary will move and change. In an open system, this balance is not necessary anymore. That is why we cannot consider the vacancy part alone, and we bring interstitials into the discussion. We know two facts about the interstitial part. One is that interstitials will move from the bulk to the grain boundary. The other is that interstitials diffuse faster than vacancies. With irradiation, more atoms, both Ni and Cr, arrive to the grain boundary as interstitials. These interstitials do not need to wait for vacancies to arrive and annihilate. In other words, the interstitials become independent from vacancies except for their generation. Note that the voids still develop, and the salt still penetrates into the sample under irradiation. Radiation did not change the end result of corrosion attack. It just delayed it.

We also observed Ni enrichment and Cr depletion in the corrosion-only region, in the absence of irradiation. This observation supports corrosion-induced vacancy injection. The reason we did not show the TEM and EDX results of the corrosion-only region is that we do not want to give readers impression that one can directly compare the grain boundaries, as TEM atom intensities vary with contrast, sample thickness, and grain boundary character. We think the distribution of elements in the irradiation region is more important to show. This is because it conveys two messages, i.e., that RIS is not dominant, and that Ni enrichment and Cr depletion both occur despite RIS not being dominant.

Reviewer #3

We would like to thank Reviewer #3 for the comments. We hope that by addressing the comments one by one, we are able to support our differing interpretation of our results with proof from existing literature and from our own experiments.

- Comment 1): *On page 1, Line 33 “radiation damage always results in negative effects”, this statement is not right, some experiments have shown that radiation can have some positive effects on corrosion, even for neutron radiation.*

We agree that some experiments have shown that radiation can have some positive effects on materials. In fact, on page 3, line 45, we wrote *In a more general sense, radiation has been noted to improve mechanical properties of structural materials in certain circumstances.* However, for current reactors, in most of the cases, radiation will accelerate corrosion. This is mentioned as referenced on page 1, line 35: *It has been well established that radiation accelerates corrosion of structural materials in today’s reactors, as in-core experiments and accelerator studies have shown.* The original sentence with the word “always” might be too strong. But it is rather an issue of choosing a more appropriate adverb. We are happy to modify the sentence as following: *..., challenging our view that radiation damage usually results in negative effects.*

- Comment 2): *On page 1, Line 52 “making corrosion in molten salt different from aqueous corrosion” actually, both types of corrosion in molten salt and aqueous solution are electrochemical process including anodic dissolution and cathodic reduction, the corrosion mechanisms are the same.*

We were not expressing that molten salt corrosion is not an electrochemical process. We copy the

relevant sentence here: *Since corrosion products and normally-protective oxides are soluble in the salt, oxide-based passivation does not occur, making corrosion in molten salt different from aqueous corrosion.* The first half of the sentence expresses that since molten salt corrosion does not result in a passivation layer due to oxide solubility in fluoride salt, it is indeed fundamentally different from a typical aqueous corrosion mechanism, despite still being an electrochemical process. We feel that the way we wrote the sentence expresses what we meant to say. Therefore, we have not modified the original manuscript based on this comment.

- Comment 3): *On page 1, Line 53 “the predominant... , is selective dissolution of Cr” Actually, selective dissolution of Cr is just one type of the corrosion, which depends on the corrosion potential of the molten salt. if the corrosion potential is above the Ni dissolution potential, Ni can be dissolved very faster than Cr. Corrosion not only depends on the thermodynamic potential but also depends on the dissolution kinetics when the corrosion potential is above its dissolution potential.*

We agree with the reviewer’s logic here, though our discussion was based on a practical material facing molten fluorides. Neither our discussion nor our experiments belong to the case where the more inert element in the alloy (Ni in our case) is also vulnerable to attack. In other words, our results are restricted to the physical situation where one element is noble and one is susceptible to corrosion, which appears in most practical applications of structural materials in molten fluoride salts. Nonetheless, to make our description more precise, we have changed the original sentences to *The predominant mechanism of corrosion of Ni-based alloys in molten fluoride salts in a practically useful sense, such as the Ni-20Cr alloys in this study, is selective dissolution of the Cr (here the most susceptible element) into the salt.*

- Comment 4): *On page 1, Line 69-71, “in liquid lead... oxidation was the primary corrosion mechanism”. Actually, in liquid lead, the major corrosion mechanism is physical dissolution, not oxidation, the oxygen control technology is applied in liquid lead to reduce the corrosion by stopping the physical dissolution.*

We agree that oxidation is employed as a way to reduce direct physical dissolution, including in liquid lead systems, as active high oxygen control is used to force a passive layer to form. However, when the oxide is not stable, physically or chemically, oxidation itself becomes a deleterious corrosion mechanism. In liquid lead under a reducing environment, corrosion occurs mainly by physical dissolution of the more soluble element(s). If it occurs under a well-controlled oxygen environment, then oxidation will happen. Then corrosion shifts from physical dissolution to oxide formation and destruction. The reference we cited was the liquid lead corrosion and irradiation experiment. In that experiment, the liquid lead was under high oxygen control, not a reducing atmosphere. Nonetheless, we noticed that the sentence was not appropriate for another reason. In the previous sentence, we wrote: *However, we observed that proton irradiation actually decelerates corrosion in molten salt, contradicting radiation accelerated corrosion observed in liquid lead, molten salt, and water-based environments while corroborating others.* Then in the next sentence, we wrote: *In these cases, oxidation was the primary corrosion mechanism, while in our case reaction/dissolution of Cr is dominant.* We should not have wrote *In these cases*, as the previous sentence also mentioned one molten salt experiment, where the dissolution of Cr is also dominant. Moreover, we think the previous sentence, by listing the related experiments by other researchers, is enough for the message. Then this summary sentence, is not necessary anymore. Therefore, we decide to delete this sentence.

- Comment 5): *On page 3, Eq.1 seems not right, the unit on the left and right are not the same, the authors need to double check the reference and actually understand what the equation is in the reference [25].*

Equation (1) is exactly the same as the equation shown on page 399 in reference [25] except for the unitless correlation coefficient. Here that equation is copied for reference: $D_a = f_v D_v C_v + f_i D_i C_i$. The same way of expressing the equation also appears in chapter 5 of Prof. Was’s book (Was, Gary S.

Fundamentals of radiation materials science: metals and alloys. springer, 2016.). The concentrations here are site concentrations, which are unitless. Therefore, the equation is balanced in terms of its units, as all diffusivities are expressed in $\frac{m^2}{sec}$. Thanks for pointing this out. The sentences after equation (1) is modified to be more clear: *where D_i and D_v represent interstitial diffusivity and vacancy diffusivity, respectively, and C_i and C_v are site concentrations of interstitials and vacancies.*

- Comment 6): *On page 3, The authors think that it is EuF_3 that leads to corrosion. I highly doubt it. I read through the reference cited, the original reference was proposed to use Eu^{3+}/Eu^{2+} couple to mitigate corrosion by decreasing corrosion potential. The author needs to make at least a very simple thermodynamic calculation to show that the standard potential of Eu^{3+}/Eu^{2+} is more positive than Cr^{2+}/Cr , if it is more positive, which means it is possible that it is Eu^{3+} that leads to corrosion, or it is impossible*

Thanks for bringing up this discussion. We actually did the thermodynamic calculation based on the reference cited: Guo, Shaoqiang, et al. "Measurement of europium (III)/europium (II) couple in fluoride molten salt for redox control in a molten salt reactor concept." Journal of Nuclear Materials 496 (2017): 197-206. Using the data from figure 8 of the reference, one can estimate that a ratio of Eu^{3+}/Eu^{2+} being 0.15 crosses the Cr^{2+}/Cr potential at 650 °C. In other words, if the ratio of Eu^{3+}/Eu^{2+} is higher than 0.15, it will corrode Cr by fluoridation. We only added Eu^{3+} to our salt. Therefore, at the very beginning, this ratio is obviously far above 0.15. In the same reference, they measured and estimated that the ratio of Eu^{3+}/Eu^{2+} is 1.68 on average using cyclic voltammetry, and 2.16 using X-ray photoelectron spectroscopy (XPS) at 650 °C. They speculated that the ratio could be from the thermal dissociation of EuF_3 . Whether it is from thermal dissociation is out of the scope of this discussion. Nonetheless, both methods show that the ratio will be above 0.15, the threshold ratio to corrode Cr. In fact, the salt in the reference was FLiNaK with 1 wt. % EuF_3 . In our experiment, the salt contained 5 wt. % EuF_3 . No matter whether there is thermal dissociation, our salt will have a ratio of Eu^{3+}/Eu^{2+} at least higher than either 1.68 or 2.16.

We also conducted some XPS measurements of the salt after our experiments. The ratio of Eu^{3+}/Eu^{2+} is estimated to be 12 for the 2.0 $\mu A/cm^2$ experiment, and 7.8 for the 2.5 $\mu A/cm^2$ experiment. Our XPS measurements show a much higher ratio compared with the ratio reported in the reference cited. This might be caused by the higher original weight percentage of EuF_3 added into FLiNaK. Nonetheless, the ratio of Eu^{3+}/Eu^{2+} is much higher than 0.15.

Now we would like to discuss how corrosive our salt is in terms of its ability to hold Cr. A simple estimation can show that the total number of Cr atoms in the Ni-20Cr foil is $8.8 * 10^{19}$. Apart from the Ni-20Cr sample, the salt is only in contact with pure Ni in our experimental apparatus. So the Ni-20Cr sample is the only source of Cr in the experiment. If Cr becomes Cr^{2+} , that is $1.76 * 10^{20}$ elementary charges in total. In our salt, the total amount of Eu^{3+} is $5 * 10^{20}$ at the beginning. If all Cr atoms in the entire sample get attacked and enter into the salt, the ratio of Eu^{3+}/Eu^{2+} will be around 1.8, which is still higher than 0.15. Even if there is thermal dissociation suggested by the reference, we can still do the same estimation. We can assume the salt has a ratio of 2 to start, which corresponds to $3.3 * 10^{20}$ Eu^{3+} . Again if all Cr atoms in the sample get attacked and enter into the salt, the ratio of Eu^{3+}/Eu^{2+} will be around 0.88, which is still higher than 0.15. Note this is a conservative estimation. The samples still have a significant number of Cr atoms as the attack is localized, as evidenced by considerable Cr remaining in the foils in our EDX investigations.

With that, we would like to discuss the accuracy of the estimations used in the reference. The thermodynamic calculation made in the reference is based on the assumption that activity of dissolved metal ions is 10^{-6} . To our knowledge, there are no activity measurements existing for this molten salt, and the number 10^{-6} is borrowed from the field of aqueous corrosion as a default activity. Therefore, the calculation of the potentials might not be accurate. However, the results generated in the reference are of good agreement with our experimental observation. That is the reason we trust the calculation made in the reference. Using XPS to measure and calculate the ratio of Eu^{3+}/Eu^{2+} also has potential issues such as sample uniformity and peak fitting errors. That is why we did not include our XPS measurements in the manuscript.

Last, we would like to specifically respond to the point: “the original reference was proposed to use $\text{Eu}^{3+}/\text{Eu}^{2+}$ couple to mitigate corrosion by decreasing corrosion potential”. There is no disagreement between our observation and the conclusion made in the reference. The $\text{Eu}^{3+}/\text{Eu}^{2+}$ couple can certainly be used to mitigate corrosion if the ratio is controlled to be low. In fact, we would say the $\text{Eu}^{3+}/\text{Eu}^{2+}$ can be used to “control” corrosion, which is exactly what is means by “redox couple”. With a high ratio, it will lead to corrosion. A low ratio will mitigate corrosion. In our experiment, we have a higher ratio that will lead to corrosion, which is exactly what we observed. We therefore believe that the discussion here fully addresses the concerns of reviewer #3 in this comment. Therefore, no change has been made to the manuscript corresponding this comment.

- Comment 7): *The impurity of the salt is unknown, especially, the oxygen level and the moisture lever, both oxygen and moisture can be the corrosion sources. Therefore, the oxygen level and moisture level at which they prepared the salt must be given.*

Thanks for pointing this out. We agree with the face that salt preparation processes are important. That is why we wrote in part A of methods, on line 75 of page 3, *The FLiNaK salt used for this work was produced in a temperature-controlled furnace housed inside an argon atmosphere glove box. Oxygen and moisture in the glove box were controlled to remain below 1 ppm.* Since we have already wrote this, and the original sentences have already addressed the comment directly, we did not modify the original manuscript corresponding to this point.

- Comment 8-1): *Figure 4 is not complete. The authors need to know that the molten salt corrosion includes two reactions: anodic dissolution/oxidation and cathodic reduction. The corrosion of Cr is through a reaction $\text{Cr} - 2e \rightarrow \text{Cr}^{2+}$, there must be a species that can be reduced to “eat” electron released by Cr dissolution. The species must be diffuse in the salt to reach the corrosion surface, the second reaction was completely neglected in Figure 4. I do not think the author really understand the corrosion mechanisms by molten salt.*

Thanks for explaining the corrosion mechanism. To answer this concern we draw focus to part A of our methods, on line 90 of page 3, where we specifically mention the cathodic reaction. From the response to comment (6), we have also mentioned the proof of proposed cathodic reaction. We did not show what happens to Cr in the salt, but not because we don’t understand the corrosion mechanism. The reason is that the goal of this manuscript, especially figure 4, is not to discuss the specific corrosion reaction(s) occurring in our molten salt. We also call specific attention to the caption of Figure 4, which is copied here for reference: *(a) Schematic of the solid state diffusion processes during molten salt corrosion in Ni-20Cr. (b) Schematic of the solid state diffusion processes during molten salt corrosion in Ni-20Cr under the influence of proton irradiation.* The caption mentioned clearly that the schematics are showing the “solid state diffusion processes”. Here we would like to emphasize that the purpose of figure 4 is to show how the introduced defects influence the corrosion. All focus is given to the solid phase, and none to the salt reactions which we believe are out of scope of this paper — they are not necessary to demonstrate that irradiation indeed decelerates corrosion. It is not meant to be a comprehensive schematic to show everything happens in the system. Thus, we did not change our manuscript with respect to this comment.

- Comment 8-2): *Another important factor that will lead to corrosion was not addressed in the document. The author used proton (H^+) which is completely stopped in the salt and accumulate in the salt. H^+ is a corrosive species in salt because $\text{H}^+ + e \rightarrow \text{H}_2$ which can be reduction reaction that may lead to corrosion, which is not related to radiation. It is because the author artificially introduced corrosive species into their system. Therefore, H^+ influence on corrosion during experiments needs to addressed and discussed.*

We actually partially agree with the speculation that reviewer #3 made here. In fact, we wrote on line 50 of page 3, *In our experiments, most protons stop in the salt, so the effect of injected hydrogen interstitials in the foil is negligible, though their effect on salt chemistry is strong. Unlike proton*

beams, neutrons will not introduce charged hydrogen when they stop in the salt. These two sentences have been modified based on comment (3) of reviewer #2 (please see above). We therefore believe it more appropriate to base our discussion on the revised sentences, *In our experiments, most protons stop in the salt, so the effect of injected hydrogen interstitials in the foil is negligible, though their effect on salt corrosiveness is strong. Unlike proton beams, neutrons do not interact with electrons directly and are not charged species themselves. Thus, we expect that salt undergoing neutron irradiation may be less corrosive than that undergoing proton irradiation, as neutron irradiation does not add charged chemical species to the salt (save for a small amount from transmutation production of tritium).*

From the result of the pure Fe foil, we showed the interaction between protons and the salt was making the salt more corrosive. When we were writing the manuscript, we also thought that the charged hydrogen might be contributing to the intensified corrosion of the pure Fe sample. But now we do not think the direct chemical interaction, i.e., protons being charged hydrogen ions, is dominating. Through some quick estimation, we can show that the total numbers of charged hydrogen ions introduced to the salt for a 4 hour irradiation with a beam current density of $2.5 \mu\text{A}/\text{cm}^2$, are $4.5 * 10^{16}$. The total numbers of Eu^{3+} are $3.5 * 10^{20}$. Thus, the direct chemical influence might be suppressed by the Eu^{3+} in the salt. However, the protons still carry kinetic energy when they enter the salt. The kinetic energy is going to dissipate mainly via electronic interactions. The species generated might be short-lived, but they likely still contribute to corrosion in a process akin to radiolysis. Additionally, there might also be influence on the anodic reaction itself, or on the surface electrical field, where we do not have enough knowledge on which to comment. In short, the real interaction, whether it is direct chemical, or chemical after physical, is not clear. Correspondingly, we remained very careful when introducing the results of the pure Fe foil. We only wanted to show that the interaction between proton irradiation and salt in our system was making the salt more corrosive. We would like to focus our scope only on the influence of the proton irradiation on the solid state diffusion process. Therefore, it is out of our scope and knowledge to investigate the influence of charged hydrogen on corrosion in molten FLiNaK. Apart from the changes already made, no other modification was performed as a result of this comment.

Reviewers' Comments:

Reviewer #2:

Remarks to the Author:

The author satisfactorily addressed all of my comments and made appropriate changes to the manuscript.

Reviewer #3:

Remarks to the Author:

All my comments are well addressed, therefore, I recommend publishing the paper

NATURE COMMUNICATIONS

Proton Irradiation-Decelerated Intergranular Corrosion of Ni-Cr Alloys in Molten Salt

W.Y. Zhou, Y. Yang, G. Q. Zheng, K. B. Woller, P. W. Stahle, A. M. Minor, M. P. Short

June 3, 2020

To the Editors and Reviewers of Nature Communications,

Thank you for reviewing our manuscript entitled "Proton Irradiation-Decelerated Intergranular Corrosion of Ni-Cr Alloys in Molten Salt."

Reviewer #2

Comment: *The author satisfactorily addressed all of my comments and made appropriate changes to the manuscript.*

We would like to thank Reviewer #2 once again for the insightful comments leading to the revised manuscript.

Reviewer #3

Comment: *All my comments are well addressed, therefore, I recommend publishing the paper.*

We greatly appreciate the comments from Reviewer #3 to help us improve the manuscript.